

# A hybrid model based on CNN-LSTM for assessing the risk of increasing claims in insurance companies

Walaa Gamaleldin[1,2,*], Osama Attayyib[1,2,*], Mrim M. Alnfiai[3],
Faiz Abdullah Alotaibi[4] and Ruixing Ming[1]

[1] School of Statistics and Mathematics, Zhejiang Gongshang University, Hangzhou, China
[2] Department of Quantitative Methods, Faculty of Commerce, Sohag University, Sohag, Egypt
[3] Department of Information Technology, College of Computers and Information Technology, Taif University, Taif, Saudi Arabia
[4] Department of Information Science, College of Humanities and Social Sciences, King Saud University, Riyadh, Saudi Arabia
* These authors contributed equally to this work.

## ABSTRACT

This article proposes a hybrid model to assist insurance companies accurately assess the risk of increasing claims for their premiums. The model integrates long short-term memory (LSTM) networks and convolutional neural networks (CNN) to analyze historical claim data and identify emerging risk trends. We analyzed data obtained from insurance companies and found that the hybrid CNN-LSTM model outperforms standalone models in accurately assessing and categorizing risk levels. The proposed CNN-LSTM model achieved an accuracy of 98.5%, outperforming the standalone CNN (95.8%) and LSTM (92.6%). We implemented 10-fold cross-validation to ensure robustness, confirming consistent performance across different data splits. Furthermore, we validated the model on an external dataset to assess its generalizability. The results demonstrate that the model effectively classifies insurance risks in different market environments, highlighting its potential for real-world applications. Our study contributes to the insurance industry by providing valuable insights for effective risk management strategies and highlights the model's broader applicability in global insurance markets.

# INTRODUCTION

The increase in insurance claims is a critical concern for insurance companies. Precisely predicting claims is not only a challenge but a necessity. The foundation of insurance operations relies on balancing risk assessment with financial stability. However, the potential for a rise in claims poses a significant threat, causing uncertainty about the ability of insurance companies to sustain their resilience and stability. The rapid and exponential growth of claims presents a considerable challenge, endangering the delicate balance between effectively managing risks and maintaining financial stability.

Corresponding authors
Walaa Gamaleldin,
walaagamal30@gmail.com
Ruixing Ming, rxming@zjgsu.edu.cn

Studying the impact of premium collections and claim payouts by insurance companies is a crucial factor that affects decision-making. Insurance companies meticulously determine insurance service prices, fulfilling specific criteria. These include ensuring that premiums are sufficient to cover expected losses and extra expenses, that pricing is profitable to allow for a margin to cover the costs of maintaining capital reserves for expected losses, and that pricing is fair to promote competition (*Parodi, 2023*). In recent years, various approaches have been proposed to improve the accuracy of risk prediction and premium estimation in the insurance industry. One notable example is the adaptive mean (ADM) method, which combines the strengths of trimming and winsorization to minimize mean square error (MSE) and enhance premium accuracy. This method significantly contributes to mitigating risk exposure caused by inaccurate premium estimations by offering a more reliable and robust model for credibility premium estimation (*Attayyib et al., 2024*).

Guaranteeing the restoration of the insured amounts to their initial state without any increase or decrease is the main principle behind the claims process. According to the insurance policy's provisions, the insurance company is responsible for providing the policyholder with enough money for compensation, repairs, replacements, or other appropriate solutions. The claim payment aims to help policyholders recover from unexpected setbacks and restore their financial stability, per the insurance coverage's purpose. Insurance claim prediction is crucial for insurers to decide which insurance plans to provide to policyholders. Insurance companies may suffer financial losses due to overpricing or underpricing resulting from inaccurate claim estimations. We need to create reliable predictive models and methodologies for anticipating claims to enhance the effectiveness and productivity of insurance operations (*Fauzan & Murfi, 2018*).

Several significant studies have explored the application of machine learning methods to predicting insurance claims. *Abdelhadi, Elbahnasy & Abdelsalam (2020)* developed an insurance claim prediction model using XGBoost, Naive Bayes classifiers, decision trees, and artificial neural networks. Their findings indicated that decision trees and XGBoost achieved the highest prediction accuracy at 92.22% and 92.53%, respectively. *Pesantez-Narvaez, Guillen & Alcañiz (2019)* compared the predictive capabilities of XGBoost and logistic regression for accident claims prediction with limited training data. Their study highlighted the interpretability of logistic regression over XGBoost, which requires extensive effort to interpret complex relationships.

Additionally, *Goundar et al. (2020)* developed an artificial neural network model for forecasting medical insurance claims, demonstrating the superiority of recurrent networks over traditional feed-forward architectures. While these models are effective, they primarily focus on static feature relationships and fail to capture insurance claims data's temporal and spatial complexities. These traditional models do not effectively model the sequential and time-dependent nature of claims, which is crucial for accurate risk prediction.

A growing body of research has leveraged deep learning techniques for insurance risk assessment, particularly convolutional neural networks (CNN) and long short-term memory (LSTM) networks. *Abakarim, Lahby & Attioui (2023)* proposed a CNN-based

fraud detection model with ensemble bagging that achieved 98% accuracy. Similarly, *Saputro, Murfi & Nurrohmah (2019)* demonstrated that combining deep neural networks with LSTM outperformed traditional feed-forward networks for auto insurance prediction. *Reddy et al. (2023)* utilized a multi-contextual modeling approach integrating CNN and bidirectional LSTM for financial fraud detection, effectively capturing spatial and sequential dependencies. Furthermore, *Lai et al. (2022)* employed LSTM to analyze brain injury insurance claims. It has achieved a notable accuracy of 74.33%. *Xia, Zhou & Zhang (2022)* introduced a joint model that combines the strengths of LSTM and CNN by extracting enhanced abstract representations, significantly improving risk assessment while reducing the reliance on domain expertise for feature engineering.

Despite these advancements, existing models exhibit notable limitations. Standalone CNNs effectively capture spatial relationships but struggle with sequential dependencies, making them inadequate for analyzing time-dependent claim patterns. Conversely, LSTM networks excel in processing sequential data but cannot capture spatial correlations inherent in insurance datasets. Traditional machine learning models, such as random forest and XGBoost, efficiently identify complex relationships but are not designed to model the temporal dependencies and intricate sequential patterns in time-series insurance data. Consequently, these methods fail to address the full complexity of insurance claims, particularly in understanding the evolving risk dynamics and spatial dependencies between different features.

To bridge this research gap, we propose a hybrid CNN-LSTM model that integrates the strengths of both architectures. By leveraging CNNs for spatial feature extraction and LSTMs for capturing temporal dependencies, our model provides a comprehensive solution for risk assessment in insurance claims. This hybrid approach enables a more robust and accurate prediction framework by effectively handling the multifaceted nature of insurance data. The proposed model outperforms conventional techniques by offering a unified solution that accommodates both spatial and temporal complexities, ultimately enhancing the reliability and accuracy of insurance risk evaluation.

Generally, insurance companies face a pressing issue with increased claims, demanding immediate attention. When the number of claims exceeds the premium revenues, it leads to financial deterioration and impairs the company's capacity to fulfill its financial responsibilities; furthermore, growing costs can reduce total profitability, necessitating additional cash through borrowing or exploring alternative financing solutions. As a result, this can affect the company's capital structure and financial liquidity. To tackle this issue, it is essential to integrate skilled risk management and meticulously adjust prices and policies to guarantee the organization's long-term sustainability and profitability.

In response to these strong reasons, this study proposes a novel method for predicting insurance claims and assessing the risk of increasing claims for premiums. Our risk model predicts claims using an innovative approach that combines LSTM and CNN capabilities. Our risk model integrates LSTM and CNN architectures to create a complete framework that appropriately evaluates the likelihood of insurance claims escalation. This technique improves our understanding of the elements that increase the possibility of insurance claims. Our method effectively analyzes the data for complex patterns and relationships,

allowing a more accurate risk assessment. It uses the temporal dependencies that LSTM networks collect and the spatial information that CNN recovers. By combining LSTM and CNN networks, our model can effectively capture the temporal and geographical characteristics of the data, making it ideal for studying the temporal evolution of insurance claims.

We select CNNs for their autonomous acquisition of spatial hierarchies and local features from data. This is especially beneficial in time series analysis, where localized patterns may signify important trends. We choose LSTM for its ability to retain information over extended durations, thereby mitigating the vanishing gradient issue prevalent in recurrent neural networks. This is essential for comprehending sequences in time-series data. We implement dropout regularization to alleviate overfitting, a prevalent challenge in deep learning models, particularly when utilizing limited datasets.

Our goal is to offer insurance companies a powerful tool for evaluating and making decisions about risk. This tool will assist them in managing and mitigating the adverse effects of increasing claims on their financial stability and profitability. Our approach enables insurance companies to make well-informed decisions about pricing, policy revisions, and risk management strategies by correctly predicting and evaluating the likelihood of an increase in insurance claims. This helps protect the organization's long-term profitability and financial stability amid increasing claims-related issues.

The main contributions of this article are as follows: (i) Create a new, useful risk model that evaluates and rates the risk factors connected with increased claims; (ii) Use a hybrid CNN-LSTM to improve the precision and dependability of risk evaluation; and (iii) By spotting patterns in historical data, our model predicts possible increases in claims, giving us useful information. Furthermore, our framework provides practical insights to improve profitability and long-term financial stability, optimize pricing frameworks, and refine underwriting procedures.

The remainder of this article is as follows. "Problem description" provides a problem description. "Methodology" provides a detailed explanation of the methodology and the proposed model. "Results and Discussions" presents the outcomes of numerical examples, illustrating how the suggested model assesses the risk of insurance companies' claims surpassing premiums and employs the optimization model to identify the optimal claims. The article concludes in "Conclusions", summarizing the essential findings and emphasizing the practical implications of our research for the insurance industry, particularly in improving profitability and long-term financial stability, optimizing pricing frameworks, and refining underwriting procedures.

## PROBLEM DESCRIPTION

Risk classification is not just a task; it is a mission-critical function in the insurance industry. It is pivotal in evaluating and controlling various risks, including the likelihood of claims exceeding premiums. We aim to categorize insurance risk into predetermined categories, such as low, normal, and high. We use historical data on insurance premiums, claims, loss rates, commission rates, administrative spending rates, and other relevant

variables. This categorization empowers insurance companies to make informed choices about risk assessment, pricing, and management, safeguarding their financial health.

To establish a precise definition of the problem, we use the following notation to develop a predictive model that effectively measures the risk of claims exceeding premiums. These models will use cutting-edge machine learning methods, like CNN and LSTM networks, to look for patterns in the time series data and find possible risk factors that affect the trends seen. $X$ represents the input feature matrix, where each row corresponds to an observation of a time series, and each column represents a distinct feature.

The dependent variable, $y$, represents evaluating the potential risk associated with increased claims compared to premiums. We can classify the variable as low, medium, or high risk.

$n$ represents the dataset's overall count of insurance companies. $m$ denotes the number of features, which is six. $t$ Denotes the total number of time points in the time series data.

To address this crucial problem, we propose using a hybrid model that combines the strengths of CNNs, which excel in spatial analysis, with LSTM networks, which are adept at temporal modeling. This powerful combination allows the model to capture and analyze intricate patterns and correlations in claims data over time, significantly enhancing our ability to manage and mitigate insurance risks.

The hybrid CNN-LSTM model is a game-changer, revolutionizing pricing decisions, underwriting processes, and risk assessment and management. It provides a practical solution to the challenges associated with rising claims, enhancing insurers' understanding of the risk landscape by capturing claims data's short- and long-term relationships. With this model, insurers can adjust premium rates and allocate reserves precisely, ensuring profitability while maintaining financial stability and sustainability in the insurance sector.

## METHODOLOGY

This section discusses the research method used to develop the prediction models. Our process is significant because it allows us to accurately predict future trends in the insurance industry, which is crucial for making informed decisions and implementing strategies to ensure the financial stability and sustainability of the insurance market.

### Data collection

We have carefully collected data from the Financial Regulatory Authority's annual reports covering the Egyptian insurance sector. This data includes details about the performance of insurance companies. This data provides accurate and comprehensive information about the performance of insurance companies, including the development of insurance premiums and claims over the past years. The data is not just numbers; rather, it carries vital insights that reflect the insurance industry's dynamics and the changes that have occurred in it. It highlights the importance of monitoring and analyzing premium and claim trends to assess insurance companies' financial performance and risk exposure.

This study includes a set of variables, including:

1. Insurance premiums ($X_1$): refer to the total amounts policyholders pay to insurance companies.

$$X_1 = \{x_{11}, x_{12}, \ldots, x_{1n}\}. \tag{1}$$

2. Claims ($X_2$): refer to the total sum insurance companies have paid to cover the submitted claims, indicating the potential financial risk they may face.

$$X_2 = \{x_{21}, x_{22}, \ldots, x_{2n}\}. \tag{2}$$

3. Loss ratio ($X_3$): Calculated as the ratio of claims paid to premiums written, it is considered a key measure of financial performance.

$$X_3 = \{x_{31}, x_{32}, \ldots, x_{3n}\}. \tag{3}$$

4. Commission rate ($X_4$): reflect the percentage of commissions paid to agents compared to total premiums written.

$$X_4 = \{x_{41}, x_{42}, \ldots, x_{4n}\}. \tag{4}$$

5. Production costs ($X_5$): include operating costs associated with underwriting and issuing documents.

$$X_5 = \{x_{51}, x_{52}, \ldots, x_{5n}\}. \tag{5}$$

6. General and administrative expenses rate ($X_6$): expresses the ratio of administrative expenses to the total subscribed premiums.

$$X_6 = \{x_{61}, x_{62}, \ldots, x_{6n}\}. \tag{6}$$

7. Level of risk ($Y$): classifies the level of risk into three categories: low, normal, and high.

$$Y = \{y_1, y_2, \ldots, y_n\}. \tag{7}$$

Figure S1 presents data on premiums and claims in the insurance industry over several years, from 1994 to 2022. From the data, several observations can be made. Firstly, a general upward trend in premiums and claims over the years indicates growth in the insurance market. However, it is important to note that the growth rates vary yearly. Between 1994 and 2022, premiums increased steadily, with a notable surge in growth in certain years, such as 2000, 2002, 2005, and 2012. These surges could be attributed to increased policy sales, expanded coverage, or changes in market conditions during those periods, providing insights into the insurance market dynamics.

Similarly, claims also experienced fluctuations, with both positive and negative growth rates. Several factors, such as changes in the number and severity of covered events and changes in the general state of the economy, might impact claim growth.

Notably, there are instances where the growth rates of premiums and claims diverge. For example, in certain years like 2001, 2004, and 2013, while premiums continued to rise, claims experienced negative growth rates. This suggests implementing effective risk management practices or relatively fewer covered losses during those periods, which could result from strategic decisions made by insurance companies.

The loss ratio, a crucial metric in evaluating an insurance company's performance, is calculated by comparing claims paid over a specific period to premiums collected during the same period. A low loss ratio indicates strong performance, while a high loss ratio suggests higher risks and may necessitate improved management and assessment procedures to control claims and enhance the company's financial sustainability.

Figure S2 represents the insurance industry's loss ratio from 1994 to 2022. The loss ratio varied between 0.40 and 0.87 during the study period. Noteworthy peaks in the loss ratio were observed in 2002, 2006, and 2012, indicating periods of relatively higher claims than premiums collected. The lowest loss ratio was recorded in 2020 at 0.40. These findings suggest fluctuations in risk exposure and the potential challenges insurance companies face in managing and covering claims over time.

## Data description

The dataset, a crucial foundation of our article, is a comprehensive information collection from 36 insurance companies. It considers each company's unique factors. The input variables considered are premiums ($x_1$), claims ($x_2$), loss rate ($x_3$), commission rate ($x_4$), production costs ($x_5$), and general and administrative expenses rate ($x_6$). The output variable ($y$) classifies the level of risk into three categories: low, normal, and high.

We use a fusion process to select the ultimate fusion value from the pooled data with the highest prediction rate. This method ensures that the fusion output captures either classifier's most reliable predictions. Our technique combines LSTM and CNN models to enhance classification performance and provide more precise and reliable results across various tasks. The goal is to accurately classify new risk classifications into one of three categories: low, normal, or high. Next, let's look at a collection of claims inputs.

$$X = \begin{pmatrix} x_1^1 & \cdots & x_1^t \\ \vdots & \cdots & \vdots \\ x_{36}^1 & \cdots & x_{36}^t \end{pmatrix} = \left( x^1, \ldots, x^i \ldots, x^t \right), \tag{8}$$

where the vector $x^i = \left( x_1^i, \ldots, x_{36}^i \right)^T$ represents the input from a set of 36 companies at a particular time $i$, each comprising $t$ samples.

## Data preprocessing and model architecture

This section describes the steps taken to prepare the input data in detail. These steps were essential in ensuring the incoming data would work with the CNN and LSTM model, which is an integral part of our research. The data preprocessing phase is not just a formality but a vital step that directly influences the performance and accuracy of the selected model architecture. Data preprocessing involves preparing and cleaning raw data before using it for analysis or modeling.

We meticulously and methodically divided the data into two distinct components: features ($X$) and target labels ($y$). The features encompass a variety of factors, including premiums, claims, loss rates, commission rates, production costs, and general and administrative expenses rates. These features serve as the basis for our model's predictions. On the other hand, the target outputs classify the level of risk into three categories: low, normal, and high. This categorization is crucial for the model's learning process.

Before applying preprocessing techniques, all personally identifiable information was removed to maintain data privacy and compliance with ethical guidelines. Additionally, we incorporated an external dataset (Health Insurance Dataset India) to reduce potential biases for model validation, ensuring its applicability beyond the Egyptian insurance sector.

In addition to ensuring data privacy, we applied rigorous preprocessing techniques to enhance data quality and reliability. Missing values were handled using multiple imputation methods, ensuring data completeness across all features. We also detected and addressed outliers using the Interquartile Range (IQR) method and Mahalanobis distance. Identified outliers were either transformed or removed based on statistical thresholds to prevent distortions in model training. These steps improved the dataset's quality, improving model stability and robustness.

A critical aspect of evaluating model performance is identifying potential biases in the dataset. The FRA dataset primarily represents the Egyptian insurance sector, where underwriting and claims settlement practices may differ from those in other markets. Additionally, variations in premium structures, claims patterns, and expense ratios across different companies could introduce biases in model predictions.

To mitigate these biases, we applied data normalization and scaling techniques to standardize input variables, ensuring that no single feature dominates the learning process. Furthermore, we incorporated an external dataset to assess the model's generalizability beyond the FRA dataset (Health Insurance Dataset India). This additional validation helps determine whether the proposed model can effectively classify risks in different market environments, reducing the likelihood of overfitting to a single dataset.

The dataset was then divided into training and testing sets using a 90–10 split, ensuring a reliable assessment of model performance. To further validate model robustness, we applied 10-fold cross-validation, which systematically tests performance across different data splits, confirming the model's consistency and reliability.

Before inputting the data into the CNN-LSTM model, we conducted preprocessing procedures to ensure compatibility with the selected architecture. We transformed the data to align with the CNN input structure, treating each feature as an independent channel. We converted the input data into a three-dimensional tensor with dimensions (batch_size, num_features, num_samples). Here, batch_size refers to the number of samples, num_features represents the number of features, and num_samples indicates the number of time steps.

We can summarize the most crucial steps for implementing the proposed model as follows.

I. Data preparation: Collect historical data regarding claims and premiums from the insurance companies and partition the data into training and testing sets.

II. Feature extraction: Identify and extract relevant aspects from the data to aid in predicting future claim risk. Variables may encompass the general and administrative expenses rate (GAE), commission rates (CR), production costs (PC), and loss rate (LR).

III. Data preprocessing: Normalize or scale the input features to standardize their scale. This stage is crucial for the model to learn efficiently.

IV. Model architecture: Create the architecture for the CNN-LSTM model. CNN layers extract spatial information from incoming data, while LSTM layers capture temporal dependencies and patterns.

V. Train the CNN-LSTM model with the provided training data. We analyze the input features to forecast future claim risk and optimize the model's performance by adjusting hyperparameters such as learning rate, activation function, loss function, dropout, epoch, batch size, and optimizer.

VI. Evaluate the trained model using the testing data. To evaluate the model's performance, compute key metrics like accuracy, precision, recall, mean squared error, and the F1 score to ensure effectiveness in measuring the risk of increasing claims.

VII. Risk assessment: After training and evaluating the model, we can use it to forecast the likelihood of increasing claims for premiums in real-world situations. Input the necessary features, and the model will generate a risk score or probability representing the chance of future claims.

VIII. Monitor and improve the model's performance continuously, as necessary, by periodically retraining the model with new data to maintain its accuracy and relevance.

## Proposed model

It is crucial to develop a model that predicts claims risk. To build this model, we use historical data that serves as a reference for past claims. We use this data as training data to classify claim risk, uncovering patterns and trends that facilitate predicting future claims. To increase prediction accuracy, we use both CNN and LSTM approaches. We use CNN for deep feature extraction and LSTM for sequence prediction based on the retrieved features. Below is a brief explanation of each model.

### CNN model

One of the most effective deep learning tools is CNN, extensively utilized in image and video recognition, natural language processing, pattern recognition, and feature extraction (*Cui, Chen & Chen, 2016*; *Shrestha & Mahmood, 2019*). The main idea of CNN is that it can take inputs at the top layer and extract local features, then transport those elements down to deeper levels to create more complex features (*Islam, Islam & Asraf, 2020*). Regarding structure, CNN is a feed-forward neural network design

combining deep structures, such as convolution, pooling, and fully connected layers (*Moradzadeh et al., 2021*) (See Fig. S3). Each layer in the CNN framework serves the following purposes.

I. The convolutional layer uses a few filters built into its structure to perform the feature extraction process. Convolutional layers possess a multitude of kernels and parameters. Every kernel corresponds to the entire input depth and has its permissible fields. These filters can recognize regional patterns and characteristics (*LeCun, Bengio & Hinton, 2015*; *Lundervold & Lundervold, 2019*). The Rectified Linear Unit (ReLU) activation function is used to execute the convolution operation on each layer as follows

$$f(x) = \begin{cases} 1 \ if \ x > 0; \\ 0 \ otherwise. \end{cases} \tag{9}$$

The available filters create the feature map using an activation function procedure in the manner described below

$$z_j^l = f \left( \sum_{i \epsilon M_j} x_i^{l-1} * w_{ij}^l + b_j^l \right), \tag{10}$$

where $z_j^l$ is the output of the l-th filter in the convolutional layer j; f shows a nonlinear function, operator $*$ presents convolution; $w_{ij}^l$ is the convolutional kernel in the l-th layers between the i-th input and the j-th output maps; $M_j$ Denotes the collection of interconnected indices corresponding to the neurons in the preceding layer; and $b_j^l$ is bias.

II. The pooling layer efficiently handles the issue of creating a new feature map. The dense packing of these feature maps makes them too large for computational processing. The pooling layer uses a downsampling procedure to calculate feature map, reducing duplicate features while maintaining noticeable characteristics (*Scherer, Müller & Behnke, 2010*; *LeCun, Bengio & Hinton, 2015*; *Moradzadeh et al., 2021*), as follows

$$z_j'^l = f \left( \sum_{i \epsilon M_j} down\left( x_j^{l-1} \right) + b_j^l \right), \tag{11}$$

where downsampling down () can be a max pooling operation.

III. A fully connected layer is a feedforward neural network with interconnected neurons. The features that the convolution and pooling layers extract are input to fully connected layers. Fully connected layers then compute the weights and biases for the features, classifying them in the final layer of CNN. Each neuron in a fully connected layer receives input values and translates them into a single output value as per the following equation (*Yamashita et al., 2018*; *Shrestha & Mahmood, 2019*)

$$z = f \left( \sum_{i=1}^{n} k_i x_i - b \right), \tag{12}$$

where $z$ represents the out value, $k_i$ is the neuron's sensitivity to input values, and $b$ is the bias.

### LSTM model

LSTM is a powerful deep learning method that stands out for its ability to store information in both the short and long-term (*Tang et al., 2023*). It was created to address the vanishing gradient problem of traditional RNNs and enhance the capture of long-range dependencies in sequential data. Storing and retrieving information over extended periods enables the network to identify and learn patterns in the input sequence (*Zhao et al., 2016*). Three key components, the forget gate, input gate, and output gate, form the foundation of the LSTM design (*Graves, Mohamed & Hinton, 2013*) (see Fig. S4).

The forget gate, a key element in the overall LSTM design, is crucial in determining what needs to be removed from the current storage block and what can be kept for future transfers. It also plays a role in managing data from the previous storage block. Feed-forward and sigmoid layers implement the 'forget gate' function. Using the sigmoid function, the forget gate generates an output ranging from 0 to 1 based on the current unit input $l_t$ and the previous unit output $h_{t-1}$. A value of $f_t$ indicates the degree of information retention. A value of 0 indicates "all forgotten," while a value of 1 indicates "all retained." This role of the forget gate makes LSTM a unique and powerful deep learning method. The particular calculation formula is as follows

$$f_t = \sigma \left( W_{fl}\, l_t + W_{fh}\, h_{t-1} + b_f \right), \tag{13}$$

where $t$ is the time step, $\sigma$ is the sigmoid function, $W$ is the weight matrix, and $b_f$ is the vectors' bias matrices.

The input gate, a critical memory cell component, controls how data from the current input can reach the memory cell. This enables the network to selectively update the cell state based on the relevance of the input and the current state. It comprises two key functions: the sigmoid function and the *tanh* function. The sigmoid function is responsible for controlling the flow of information. In contrast, the *tanh* function generates a vector based on the input data, including the output of the previous unit $h_{t-1}$ and the current input $l_t$. Then, this vector is used to update the memory cell

$$i_t = \sigma(W_{il}l_t + W_{ih}h_{t-1} + b_i) \tag{14}$$

and

$$a_t = tanh(W_{al}l_t + W_{ah}h_{t-1} + b_a), \tag{15}$$

where $i_t$ and $a_t$ are the input gate's outputs, *tanh* denotes the *tanh* function, and the meanings of the other symbols are as previously explained. Based on this, the forget gate and input gate can be combined to update the cell's state

$$a_t = f_t \otimes a_{t-1} + i_t \otimes a_t, \tag{16}$$

where $a_{t-1}$ denotes the previous cell state and $a_t$ current cell state, $\otimes$ denotes the element-wise product of the vectors.

| Algorithm 1 LSTM forward propagation algorithm. |
| --- |

| | |
| --- | --- |
| 1. | **inputs:** Feature matrix $l(t)$ and target matrix $h(t)$. |
| 2. | Initialize: Randomly initialize the weight matrix W and bias vectors b. |
| 3. | Forget gate: $f_t = \sigma(W_{fl}\, l_t + W_{fh}\, h_{t-1} + b_f)$ . |
| 4. | Input gate: $i_t = \sigma(W_{il}l_t + W_{ih}h_{t-1} + b_i)$. |
| 5. | Candidate value: $a_t = tanh(W_{al}l_t + W_{ah}h_{t-1} + b_a)$. |
| 6. | Update the cell state: $a_t = f_t \otimes a_{t-1} + i_t \otimes a_t$. |
| 7. | Update the output of the LSTM:<br>$o_t = \sigma(W_{ol}l_t + W_{oh}h_{t-1} + b_o)$,<br>$h_t = o_t \otimes \tanh(a_t)$. |
| 8. | **Outputs:** The estimation value h(t). |

The output gate, another key component of the LSTM, plays a crucial role in controlling the exposure of the internal memory cells. It does this by receiving as inputs from the output of the previous cell $h_{t-1}$ and the current cell input $l_t$, then using the sigmoid function to return the output. The output of the output gate $o_t$ and the cell state $a_t$ scaled by the *tanh* function are fully considered in the final current cell output $h_t$. To illustrate this, let's walk through the calculation

$$o_t = \sigma(W_{ol}l_t + W_{oh}h_{t-1} + b_o) \tag{17}$$

and

$$h_t = o_t \otimes \tanh(a_t). \tag{18}$$

### Hybrid CNN-LSTM model

We designed our model to handle the intricacies and subtleties inherent in insurance data. Its objective is to categorize risk levels into low, normal, and high. We used the dataset's information to create a model structure that matches the data and outperforms other methods. This introduction provides an overview of the reasoning behind the creation of our suggested model, emphasizing its main characteristics and benefits.

After thoroughly examining the dataset, we have determined that a strong and flexible model is required to handle the complexities of insurance data effectively. Conventional machine learning methods frequently fail to capture insurance datasets' intricate connections and time-based patterns. Previous models, such as standalone CNNs, LSTMs, and traditional machine learning techniques like random forest or XGBoost, have demonstrated effectiveness in insurance claim prediction. However, these methods have significant limitations in addressing the spatiotemporal complexities inherent in insurance data. Standalone CNNs are excellent at detecting spatial patterns but struggle to capture temporal dependencies in sequential data. Similarly, LSTM models are adept at handling time-based data but often fail to consider spatial features crucial in insurance claims. While efficient in capturing complex relationships, traditional machine learning models like Random Forest and XGBoost are not designed to handle the temporal dependencies and

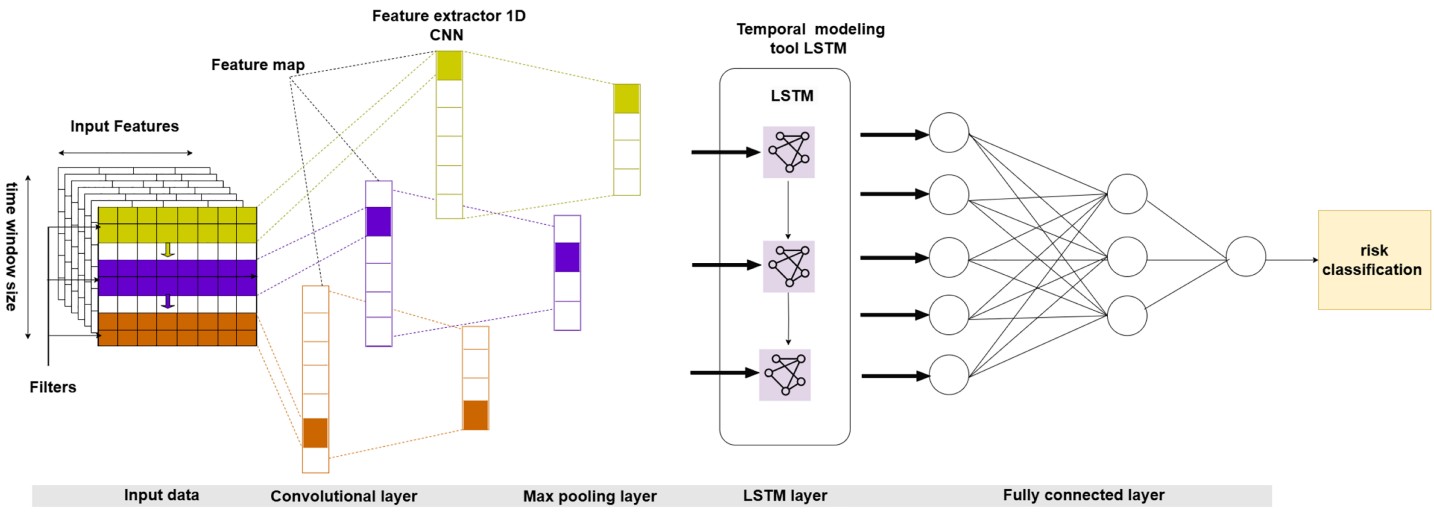

**Figure 1  Proposed architecture of a hybrid CNN-LSTM.**

intricate patterns in time-series insurance data. As a result, these methods fail to address the full scope of the complexities in insurance data, particularly in understanding the dynamics over time and the spatial dependencies between various features. To address these gaps, our proposed hybrid CNN-LSTM model combines the strengths of both architectures, effectively capturing both spatial and temporal patterns. Doing so offers a more robust and accurate approach to insurance risk assessment, providing a comprehensive solution that outperforms conventional models in handling the multifaceted nature of insurance claim data. Hence, we developed an innovative model structure that integrates the advantages of CNN and LSTM networks, referred to as the CNN-LSTM model.

We specifically tailor the suggested model to incorporate the distinctive attributes of insurance data, such as temporal dynamics, spatial interdependence, and sequential patterns. Our superior alternative model combines CNN to extract spatial features and LSTM to capture temporal relationships (*Islam, Islam & Asraf, 2020*). This integration creates a comprehensive framework for risk categorization that surpasses traditional methods. By conducting thorough experimentation and evaluating performance, we provide evidence of the usefulness and superiority of our suggested approach for appropriately categorizing risk levels (*Chen, 2016*). Figure 1 shows the broad form of the hybrid CNN-LSTM model intended for insurance claims prediction.

Our model's architecture is a fusion of CNN and LSTM layers, commonly known as the CNN-LSTM model. The CNN module of the model is not just a component but a crucial element responsible for analyzing the spatial characteristics derived from the input data. We used a Conv1D layer with a kernel size of 3 to extract these characteristics, followed by a MaxPooling1D layer to reduce dimensionality. We then transform the CNN model's output into a one-dimensional array and feed it into a fully connected layer, a crucial step in the model's learning process.

Next, we changed the CNN model's output to work with the LSTM layer's inputs. The LSTM layer is in charge of handling the data's sequential properties and capturing temporal relationships. To achieve this, we used an LSTM layer with 64 units and a rectified linear unit (ReLU) activation function. We also used a dropout layer to reduce overfitting with a rate of 0.5.

The model uses a tensor structure to represent the input data, organized as a three-dimensional tensor. Each dimension of the tensor corresponds to distinct features of the data. The first dimension represents the batch size, the second defines the number of features (channels), and the third represents the number of time steps (samples).

To ensure the robustness and reliability of the proposed CNN-LSTM model, K-fold cross-validation with 10 folds was implemented. This method systematically partitions the dataset into training and validation subsets, enabling multiple evaluations across various data splits. The model's performance was assessed in each fold using accuracy, precision, recall, and F1-score metrics, ensuring that results are not biased due to a single train-test split. The cross-validation results confirm the model's stability across different subsets of the dataset.

The chosen model structure and the data preprocessing steps are compatible and meticulously designed to be consistent. This design choice makes it easier to accurately group risk levels into low, normal, and high levels, providing a solid foundation for our research.

The following sections present a comprehensive analysis of the structure, execution, and assessment of the suggested CNN-LSTM model. We give empirical findings demonstrating the model's performance metrics, encompassing accuracy, precision, recall, F1 score, and other pertinent indicators. In addition, we analyze the consequences of our discoveries and emphasize the possible uses and advantages of the suggested model in practical insurance situations.

### Hyperparameters and model performance

This section focuses on the essential elements of hyperparameter tuning and model performance evaluation within the framework of our proposed CNN-LSTM model for risk categorization in insurance companies. Hyperparameters are crucial in defining the effectiveness and efficiency of machine learning models. They significantly impact the models' ability to generalize to new data and attain optimal performance. Similarly, model performance indicators offer vital insights into the effectiveness and dependability of the suggested approach, influencing decision-making and providing information for future enhancements.

Hyperparameters refer to a wide range of configuration options that determine the behavior and structure of the model. These settings include the learning rate, batch size, number of layers, activation functions, and regularization approaches. The process of choosing and adjusting hyperparameters is crucial. It necessitates thoughtful analysis and experimentation to find the right balance between the complexity of the model and its capacity to generalize. This section investigates the influence of various hyperparameter configurations on the performance of our CNN-LSTM model. The objective is to

determine the ideal settings that maximize predicted accuracy and resilience. The values displayed in Table S1 are the basis for using the hyperparameters for the model.

The sensitivity of model performance to hyperparameter choices has been observed, where slight changes in parameters such as learning rate or batch size can lead to significant variations in model outcomes. This highlights the importance of careful hyperparameter tuning in optimizing model performance and preventing overfitting. A detailed sensitivity analysis will explore these effects across different settings and datasets in the following sections.

In the following sections, we thoroughly examine the model's effectiveness on several datasets and experimental scenarios, emphasizing significant discoveries and their relevance to practical use in the insurance sector. This section provides a fundamental framework for comprehending the complexities of hyperparameter tuning and evaluating model performance. It sets the stage for future progress and improvements in risk classification approaches.

## Integrated model

We use historical data to create an integrated model for evaluating the potential risk associated with increasing claims for premiums from insurance companies. This data includes information on claims (C), premiums (P), general and administrative expenses (GAE), loss rates (LR), commission rates (CR), and production costs (PC). The model design combines a convolutional neural network (CNN) and long short-term memory (LSTM) for predictive modeling. Additionally, the model utilizes the Weibull distribution for risk assessment. First, we preprocess the historical data, applying normalization to standardize the variables. Next, we conduct feature engineering to transform the time series data into sequences suitable for modeling with the CNN-LSTM architecture. We train the CNN-LSTM model to predict future claims ($\hat{C}_{t+1}$) using historical data on premiums, general and administrative expenses, loss rates, commission rates, and production costs

$$\hat{C}_{t+1} = CNN - LSTM(P_t, LR_t, CR_t, PC_t, GAE_t), \tag{19}$$

subsequently, a Weibull distribution is employed to analyze the historical claims data and establish a mathematical representation of the likelihood of different claims amounts. For various claims amounts (x), the Weibull distribution gives the probability density function (PDF) (*Chaurasiya, Ahmed & Warudkar, 2018*; *Yousefi, Tsianikas & Coit, 2022*) as follows

$$f(x) = \frac{\beta}{\alpha} \left(\frac{x}{\alpha}\right)^{\beta-1} exp\left\{-\left(\frac{x}{\alpha}\right)^{\beta}\right\}, \tag{20}$$

after that, we evaluate the risk by comparing the difference between predicted and actual claims, considering the possibility that claims exceed predetermined limits. We can calculate the risk by adding up the difference between the expected ($\hat{C}_t$) and actual ($C_t$) claims and weighting it according to the probability density function (PDF) of the Weibull distribution, as presented in the following equation

$$Risk = \sum_{t=1}^{r} (\hat{C}_t - C_t) * f(\hat{C}_t); \quad t = 1, \dots, r. \tag{21}$$

This comprehensive approach offers a robust framework for evaluating and mitigating the risk associated with claims adjustments within insurance companies.

## RESULTS AND DISCUSSIONS

### Dataset preparation

The experiments use the dataset from the Financial Regulatory Authority's annual reports for the Egyptian insurance sector (FRA) (https://fra.gov.eg/%d8%a7%d9%84%d8%aa%d9%82%d8%a7%d8%b1%d9%8a%d8%b1-%d8%a7%d9%84%d8%b3%d9%86%d9%88%d9%8a%d8%a9/) for training and testing. We include the dataset to illustrate the evolution of premiums and claims data in insurance companies. Before inputting the data into the model, we conducted preprocessing procedures to ensure compatibility with the selected architecture. We transformed the data to align with the CNN input structure, treating each feature as an independent channel.

We used an additional dataset, the Health Insurance Dataset India, for external validation to address potential concerns regarding the proposed model's generalizability. This dataset, sourced from Kaggle (https://www.kaggle.com/datasets/balajiadithya/health-insurance-dataset-india-2022), represents a different insurance market, allowing us to evaluate the model's robustness across diverse demographic and economic conditions.

### Implementation details

We use the PyTorch framework to train the proposed model. The training epochs are set to 100. The Adam optimizer is used for network parameter optimization with $Beta_1 = 0.9$ and $Beta_2 = 0.999$. The initial learning rate, weight decay, and optimizer momentum are set to 0.001, 0.0005, and 0.937, respectively. The batch size is set to 64.

The experiments are conducted on a Windows operating system equipped with an NVIDIA GTX 3090 GPU. All methods are tested under the same hardware setup to ensure experimental fairness.

### Evaluation metrics

To assess the model's efficacy, we use a set of metrics specifically designed to measure classification accuracy, precision, recall, F1-score, and other vital indicators. These measurements offer a comprehensive viewpoint on the model's capacity to accurately categorize risk levels as low, normal, and high. This, in turn, aids in making well-informed decisions and developing risk management strategies inside insurance companies.

To ensure that our CNN-LSTM model does not suffer from overfitting, we applied dropout regularization to reduce model complexity and prevent excessive reliance on specific features. However, to further validate the effectiveness of dropout regularization, we conducted additional tests across multiple datasets, including the Financial Regulatory Authority (FRA) dataset and the Health Insurance Dataset India.

Additionally, to ensure the robustness and reliability of our results, we implemented 10-fold cross-validation, which systematically partitions the dataset into multiple training and validation subsets. This method helps confirm that the model's high accuracy is not due to overfitting a specific data split but reflects its genuine predictive ability across different data partitions.

Furthermore, we validated the model on an external dataset (Health Insurance Dataset India) to assess its generalizability beyond the Egyptian insurance market. The consistent classification performance across both datasets demonstrates the model's adaptability to different market environments, mitigating concerns about dataset-specific biases.

The results demonstrated that dropout regularization, in combination with cross-validation, significantly improved the model's robustness. Specifically, we observed that the model maintained stable accuracy across different validation folds, and the variance in performance metrics (*e.g.*, accuracy, F1-score) between training and validation sets was minimal.

Furthermore, when applying the model to the additional dataset, the CNN-LSTM framework exhibited consistent predictive performance, reinforcing its ability to generalize beyond the original training data. This confirms that the implemented dropout strategy effectively mitigated overfitting and enhanced model reliability.

Through extensive experimentation, we highlight both the strengths and limitations of the proposed model, offering valuable insights into areas for improvement and proposing potential avenues for future research. The performance of the suggested system is assessed using the metrics listed below (*Xie et al., 2021*).

Confusion matrix. The performance of classification on a set of test data when the true values are known is displayed in Table S2.

a) Accuracy. It is the ratio of correctly categorized samples by the model to the total number of samples for a particular test dataset.

$$Accuracy = \frac{TP + TN}{TP + TN + FP + FN}. \tag{22}$$

b) Precision. Is the number of samples the model predicts as positive and positive samples.

$$Precision = \frac{TP}{FP + TP}. \tag{23}$$

c) Recall. The recall rate represents the proportion of correctly anticipated positive samples in the dataset.

$$Recall = \frac{TP}{FN + TP}. \tag{24}$$

d) F1-score. The F1-score is consistently close to the lower precision or recall value. It represents the harmonic mean (in percentile) of precision and recall. The F1-Score performs best when the costs of false positives and false negatives vary significantly.

**Table 1 Comparison of claims prediction with other methods on Financial Regulatory Authority (FRA) dataset.**

| DL methods | CNN-LSTM | LSTM | CNN | LogisticReg | GaussianNB | SVC | K-NN | CNN-GRU | CNN-BiLSTM |
|---|---|---|---|---|---|---|---|---|---|
| Accuracy | 98.4 | 92.2 | 95.3 | 96.8 | 81.2 | 87.5 | 65.6 | 94.4 | 93 |
| Precision | 98.5 | 92.5 | 95.7 | 96.9 | 83.8 | 87.5 | 66.8 | 95.8 | 93.2 |
| Recall | 98.4 | 92.2 | 95.3 | 96.8 | 81.2 | 87.5 | 65.6 | 94.4 | 93 |
| F1-score | 98.4 | 92.2 | 95.2 | 96.8 | 80 | 87.5 | 65.9 | 94.7 | 92.6 |

$$F1 - Score = \frac{2 * Precision * Recall}{Precision + Recall} \qquad (25)$$

where TP is true positive, TN is true negative, FN is false negative, and FP is false positive.

## Numerical results

This section comprehensively analyzes the numerical findings obtained from testing and evaluating our proposed CNN-LSTM model for risk classification in insurance companies. Numerical findings are crucial for evaluating machine learning models' efficiency, dependability, and efficacy. They offer practical insights and empirical proof to enhance decision-making. Within the scope of our work, numerical results provide valuable quantitative measures to assess the predicted accuracy, reliability, and ability to apply the suggested model to various experimental circumstances and datasets.

The numerical findings reported here cover various performance indicators designed to assess different elements of the model's effectiveness and efficiency. The measures encompassed in this list are accuracy, precision, recall, and F1-score. Every indicator provides unique perspectives on various aspects of model performance, from classification accuracy to the model's ability to grasp the diversity and subtleties within the data.

The purpose of the testing dataset is validation and to get an objective evaluation of accuracy during the learning process. The CNN-LSTM model's confusion matrix result based on the dataset is displayed in Fig. S5. Table 1 compares the claims prediction for several approaches, clearly demonstrating the effectiveness of the hybrid CNN-LSTM method. The performance comparison presented in the table highlights the superior capability of the hybrid CNN-LSTM model in accurately classifying insurance risk levels. The model achieves the highest accuracy (98.4%) across all evaluation metrics on the original dataset. While other models, such as LSTM (92.2%) and CNN (95.3%), perform reasonably well, they fall short of the hybrid model's accuracy. Traditional machine learning models, such as logistic regression, Gaussian naive Bayes (NB), SVC, and k nearest neighbors (K-NN), exhibit significantly lower performance, with K-NN achieving the lowest accuracy of 65.6%. These findings underscore the importance of using advanced hybrid models like CNN-LSTM for tackling the complexities of insurance risk classification, particularly in scenarios involving diverse datasets.

The comparative analysis of hybrid deep learning models, including CNN-GRU and CNN-bidirectional long short-term memory (BiLSTM), demonstrates that CNN-LSTM remains the most effective architecture, achieving an accuracy of 98.4%, outperforming CNN-GRU (94.4%) and CNN-BiLSTM (93.0%). While CNN-GRU exhibited competitive

performance due to its simplified gating mechanism, it did not surpass CNN-LSTM in handling long-term dependencies. Similarly, CNN-BiLSTM, despite its bidirectional processing capability, did not yield significant improvements, likely due to increased computational complexity. These results reinforce the robustness of CNN-LSTM in predicting insurance risk volatility, further validating its superiority over traditional machine learning methods and alternative hybrid architectures.

The model's robustness is evaluated beyond conventional metrics by incorporating calibration analysis and cost-sensitive evaluation. Calibration analysis uses the Brier score, which measures the mean squared error between predicted probabilities and actual outcomes. The Brier scores for each class are as follows: Class 0 achieves a score of 0.0177, Class 1 scores 0.0517, and Class 2 records a score of 0.0382. The reliability diagram (Fig. S6) visually represents the alignment between predicted probabilities and observed frequencies, reinforcing the model's reliability.

Additionally, a cost analysis is performed to assess the financial impact of classification errors in the context of insurance risk. This analysis, shown in Fig. S7, evaluates the total costs associated with different decision thresholds. It highlights the trade-offs between accepting risks and minimizing potential losses, providing essential insights for decision-making in insurance risk assessment. These results affirm the hybrid model's ability to balance prediction accuracy and financial risk, providing a comprehensive approach to efficiently managing insurance risks.

While the CNN-LSTM model achieved 98.5% accuracy on the FRA dataset, testing its generalizability on external data was crucial. When evaluated on the Health Insurance Dataset India (Table S3), the model demonstrated comparable performance, confirming its robustness in different insurance market conditions. However, slight variations in accuracy suggest the presence of market-specific factors that may impact prediction reliability.

The model also showed efficient execution times, with training durations ranging from a few seconds to minutes, indicating its feasibility for practical deployment in diverse computational environments. These results underline the model's potential to enhance risk classification processes and contribute to more effective risk management strategies in insurance companies.

The proposed CNN-LSTM model can be integrated into insurance companies' risk management processes in several ways. First, it can automate risk classification during the underwriting process, enabling insurers to make faster and more data-driven decisions. Second, it can enhance fraud detection by identifying suspicious claim patterns that require further investigation. Third, insurers can leverage the model's predictive insights to optimize premium pricing strategies based on risk levels. Lastly, insurers can proactively manage their risk portfolios and develop mitigation strategies by analyzing aggregate claim trends. These applications highlight the model's potential impact on improving decision-making efficiency and financial stability in the insurance sector.

Through our rigorous testing and analysis, we aim to clarify the capabilities and constraints of our proposed CNN-LSTM model. This comprehensive approach provides vital insights into how well the model performs in various risk categories, datasets, and experimental conditions. By thoroughly analyzing the numerical outcomes, we can

identify consistent patterns, trends, and correlations that provide insight into the model's behavior and efficacy in real-world situations. Furthermore, this thoroughness in our analysis instills confidence in the model's performance and its potential to deliver reliable results.

In the following sections, we present a thorough summary of the numerical outcomes derived from our experimentation, examining significant discoveries, observations, and practical implications for the insurance sector. We aim to provide stakeholders, legislators, and industry professionals with valuable information through numerical data, enabling them to make informed decisions, enhance risk management techniques, and boost operational efficiency in insurance businesses.

## Model-based approach

The statistical technique most appropriate for use with the dataset is the Weibull distribution, which displays the Weibull density function $f(x)$ and its two parameters, the scale parameter $\alpha$ and the shape parameter $\beta$.

Many methods can be used to estimate the parameters of the Weibull distribution; however, because of its numerical stability, the maximum likelihood estimator is the most widely used (*Cohen, 1965*). Using the previously described approach, we have obtained $\beta = 316{,}341$ and $\alpha = 0.56$ for our dataset's shape and scale parameters, respectively.

## Sensitivity analysis

We explore the complexities of sensitivity analysis and its importance in assessing machine learning models' strength, consistency, and dependability, specifically about risk classification in insurance firms. Sensitivity analysis is a robust method to assess the impact of changes in input parameters, assumptions, and conditions on a model's outputs. This study provides significant information about how the model behaves and performs in diverse scenarios.

Sensitivity analysis serves a dual purpose: firstly, it uncovers pivotal elements or variables that exert a significant influence on model predictions, and secondly, it quantifies the magnitude of their impact on the desired outcomes. Sensitivity analysis enables academics and practitioners to gain a deeper understanding of the underlying mechanisms that shape a model's behavior by systematically adjusting input parameters and analyzing the resulting changes in model outputs.

Sensitivity analysis plays a pivotal role in insurance risk categorization. It aids in evaluating the robustness and reliability of prediction models employed for risk assessment and management. Through sensitivity analysis, insurance firms can identify critical risk factors, gauge their impact on the overall risk exposure, and devise proactive strategies to mitigate potential risks and uncertainties.

In this section, we do a thorough sensitivity analysis of our proposed CNN-LSTM model. Our goal is to understand how sensitive the model's predictions are to changes in input parameters, hyperparameters, and other essential elements. It allows us to understand how various factors, such as commission rates (CR) and production costs (PC), affect our proposed model's performance. This analysis also underscores the

superiority of the CNN-LSTM model over the LSTM model. We conducted the sensitivity analysis by varying these parameters by 5%, 10%, and 15% of their initial values.

Table S4 illustrates a comprehensive sensitivity analysis of CR and PC of the LSTM and CNN-LSTM models. Notably, the CNN-LSTM model consistently outperforms the LSTM model in various scenarios. For instance, with a 15% decrease in commission rates, the LSTM model shows a sensitivity score of 0.0761. In contrast, the CNN-LSTM model demonstrates a significantly lower sensitivity score of 0.0298, indicating its resilience to changes in commission rates. Similarly, with a 10% decrease, the CNN-LSTM model consistently exhibits a lower sensitivity score, further highlighting its stability and resilience. As the parameters grow by 5%, the LSTM model's sensitivity declines to 0.0241. In contrast, the CNN-LSTM model's sensitivity declines to 0.0119, showcasing its reliability and ability to deal with even minor changes.

Notably, the CNN-LSTM model demonstrates enhanced stability compared to the LSTM model, as evidenced by sensitivity scores for production costs, indicating its superior performance under these circumstances. These analyses demonstrate the increased effectiveness and durability of the CNN-LSTM model in operational situations. The improved stability of the CNN-LSTM model highlights its ability to manage complex and ever-changing datasets effectively. This makes it an attractive option for tasks that require strong performance and dependability. Our results consistently show that the CNN-LSTM model is more resistant to parameter changes than the LSTM model, as it consistently has lower sensitivity ratings. Furthermore, as the size of parameter fluctuations increases, the CNN-LSTM model's superiority becomes even more evident, showcasing its stability and efficacy in managing uncertainty. As shown in Figs. S8 and S9, these results support the CNN-LSTM model as a possible choice for predictive modeling tasks, highlighting its adaptability in real-life situations where parameters are likely to change.

This sensitivity analysis highlights the importance of careful hyperparameter tuning, as changes in input parameters like commission rates and production costs significantly affect model performance. The CNN-LSTM model's superior performance under various conditions demonstrates its ability to maintain stable predictions, even with fluctuations in key input parameters. However, this also reinforces the need for careful management of hyperparameters to ensure the model avoids overfitting or underperformance due to sensitivity to these parameters. Sensitivity analysis is an essential part of model validation and verification. It helps insurance business stakeholders make well-informed judgments, evaluate risks, and optimize risk management and decision-making strategies.

## CONCLUSIONS

This study presents an advanced hybrid deep learning framework that integrates CNN with LSTM networks to enhance the accuracy and reliability of insurance claims risk assessment. This approach effectively anticipates future claims escalation by leveraging spatial and temporal patterns within claim data, enabling insurance companies to implement proactive risk management strategies.

Grounded in real-world data from Egyptian insurance companies, our CNN-LSTM model has demonstrated its practical applicability in identifying and mitigating claim volatility risks. Through extensive benchmarking and validation, including datasets from the Financial Regulatory Authority (FRA) and the Health Insurance Dataset India, the model has outperformed traditional predictive techniques, including standalone LSTM and statistical-based approaches. The sensitivity analysis further confirmed the model's robustness across different scenarios.

The integration of CNN for feature extraction and LSTM for sequential modeling has provided a competitive advantage by enhancing pattern recognition and long-term dependency modeling in claims data. This methodological innovation represents a significant step toward data-driven decision-making in the insurance sector, facilitating improved financial stability and long-term sustainability.

While the proposed model demonstrates promising results, the successful adoption of this model in real-world insurance companies depends on overcoming challenges related to computational resources and data preprocessing. To address these challenges, we recommend integrating automated tools for data preprocessing and leveraging cloud-based platforms such as AWS, Google Cloud, or Microsoft Azure. These platforms provide scalable computational power, reducing the need for complex in-house infrastructure. Additionally, investing in training programs for technical staff will ensure smooth implementation and long-term success of the model.

The CNN-LSTM model was comprehensively evaluated using the FRA dataset and the Health Insurance India dataset. While the model effectively classifies insurance risks, further research is needed to enhance its adaptability across diverse markets. Despite its strong predictive capabilities, the CNN-LSTM model has limitations that must be acknowledged. A key limitation is potential data bias, as the FRA dataset may not fully capture global risk patterns. Although external validation was conducted, additional studies with diverse datasets are necessary. Model interpretability also remains challenging, requiring future exploration of techniques like SHapley Additive exPlanations (SHAP) and local interpretable model-agnostic explanations (LIME) to enhance transparency. Moreover, model performance is influenced by data quality, including missing values and class imbalance, necessitating advanced preprocessing methods. Finally, the model's computational complexity and hyperparameter sensitivity present accessibility challenges, particularly for smaller insurance companies with limited resources. Reducing the model's complexity and exploring optimization strategies will enhance its practical application and scalability.

### Funding

This work was supported by the Researchers Supporting Project number (RSPD2025R838), King Saud University, Riyadh, Saudi Arabia. The funders had no role in study design, data collection and analysis, decision to publish, or preparation of the manuscript.

## Grant Disclosures

The following grant information was disclosed by the authors:

King Saud University, Riyadh, Saudi Arabia: RSPD2025R838.

## Competing Interests

The authors declare there are no competing interests.

## Author Contributions

- Walaa Gamaleldin conceived and designed the experiments, performed the experiments, analyzed the data, performed the computation work, prepared figures and/or tables, authored or reviewed drafts of the article, approved the final draf, and approved the final draft.
- Osama Attayyib conceived and designed the experiments, performed the experiments, analyzed the data, performed the computation work, prepared figures and/or tables, authored or reviewed drafts of the article, approved the final draf, and approved the final draft.
- Mrim M. Alnfiai performed the experiments, authored or reviewed drafts of the article, approved the final draf, and approved the final draft.
- Faiz Abdullah Alotaibi performed the experiments, authored or reviewed drafts of the article, approved the final draf, and approved the final draft.
- Ruixing Ming performed the experiments, performed the computation work, authored or reviewed drafts of the article, approved the final draf, and approved the final draft.

## Data Availability

The dataset from the Financial Regulatory Authority's (FRA) annual reports for the Egyptian insurance sector for training and testing illustrates the evolution of premiums and claims data in insurance companies and is available at: https://fra.gov.eg/%d8%a7%d9%84%d8%aa%d9%82%d8%a7%d8%b1%d9%8a%d8%b1-%d8%a7%d9%84%d8%b3%d9%86%d9%88%d9%8a%d8%a9/

The Health Insurance Dataset India 2022 is available at Kaggle: https://www.kaggle.com/datasets/balajiadithya/health-insurance-dataset-india-2022.

## Supplemental Information

Supplemental information for this article can be found online at http://dx.doi.org/10.7717/peerj-cs.2830#supplemental-information.

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
