# Peer review of "A hybrid model based on CNN-LSTM for assessing the risk of increasing claims in insurance companies"

_PeerJ Computer Science, doi:10.7717/peerj-cs.2830_

## Round 0.1 · original submission · Major Revisions

Major revisions are required for this paper.

Reviewer 1 ·

Basic reporting

Here are the disadvantages of the article "A hybrid model based on CNN-LSTM for assessing the risk of increasing claims in insurance companies":
Disadvantages:
1. Dataset Bias:
The study relies on data from the Egyptian insurance sector. This geographic and industry-specific dataset might limit the generalizability of the findings to other markets or insurance sectors.
2. Complexity of the Model:
The hybrid CNN-LSTM model introduces computational complexity, requiring significant resources for training, which might not be feasible for smaller insurance firms.
3. Interpretability Issues:
Deep learning models, especially hybrids like CNN-LSTM, are often criticized for their lack of transparency. This may hinder decision-makers trust and practical adoption in the insurance industry.
4. Hyperparameter Sensitivity:
As highlighted in the sensitivity analysis, the model's performance depends heavily on hyperparameter tuning. This could lead to overfitting or suboptimal results if not carefully managed.
5. Potential Overfitting:
The article mentions implementing dropout regularization to prevent overfitting, but the extent to which this was effective is not thoroughly validated across diverse datasets.
6. Limited Exploration of Alternatives:
While the paper claims superiority over standalone CNN and LSTM models, it does not thoroughly benchmark against other state-of-the-art hybrid models or simpler machine learning methods like Random Forests or Gradient Boosting.
7. Practical Implementation Challenges:
The computational requirements and data preprocessing steps outlined could pose a barrier for real-world implementation in companies without technical expertise.
8. Performance Metrics Coverage:
The paper heavily emphasizes accuracy, precision, and recall but does not fully explore other critical metrics that may reflect model robustness, such as calibration or real-world cost implications.
Recommendation:
Given the paper's significant contributions to predictive modeling in the insurance sector and its novel integration of CNN and LSTM architectures, it demonstrates strong potential for acceptance. However, the authors should address the mentioned weaknesses before final acceptance to improve the paper's applicability and reliability across broader contexts.

Experimental design

no comment

Validity of the findings

no comment

Additional comments

no comment

·

Basic reporting

-

Experimental design

-

Validity of the findings

Please explore more specific research findings to show the novelty.

Additional comments

.

Reviewer 3 ·

Basic reporting

a. The manuscript references machine learning models like XGBoost, Naive Bayes, and Decision Trees (lines 63–97), but lacks a detailed comparison of their inability to address temporal and spatial complexities in insurance data. Abdelhadi, Elbahnasy, and Abdelsalam (2020) highlight that these models may not be the most suitable for this context. Please clarify why this approach was chosen.

b. Some sentences are overly wordy and could benefit from more concise phrasing. For example, "Our risk model incorporates an innovative approach to predicting claims by utilizing the capabilities of short-term memory (LSTM) networks and convolutional neural networks (CNN)" at lines 108- 110. There are occasional grammatical issues (e.g., inconsistent verb tenses). A thorough proofreading or professional editing service is recommended to ensure consistency and polish.

c.The introduction lacks a clear discussion of limitations in previous models and the gaps that this study aims to address. Add a paragraph summarizing why existing methods (e.g., standalone CNNs, LSTMs, or traditional machine learning models like Random Forest or XGBoost) are insufficient for addressing the spatiotemporal complexities in insurance data. Clearly articulate how the proposed hybrid CNN-LSTM fills these gaps and offers distinct advantages. While it mentions the limitations of conventional models (e.g., lines 399–401), a more explicit statement of the research gap is missing. This could be an area for improvement.

Experimental design

a. The proposed CNN-LSTM model achieves high accuracy (98.5%), outperforming standalone CNN (95.8%) and LSTM (92.6%) models (lines 32–33). However, cross-validation is not discussed, leaving uncertainty about the model's robustness and reliability across different data splits. The argument is well-supported by empirical evidence, such as precision, recall, and F1-score metrics (lines 502–523). Limited exploration of alternative explanations for the results or potential biases in the dataset. Please Validate robustness using cross-validation and discuss generalizability to other datasets

b. The argument is well-supported by empirical evidence, such as precision, recall, and F1-score metrics (lines 502–523). However, limited exploration of alternative explanations for the results or potential biases in the dataset. Please add a critical discussion on how the hybrid model’s design addresses specific challenges (e.g., spatiotemporal dependencies) that traditional models cannot handle.

Validity of the findings

a. The hybrid CNN-LSTM model demonstrates an impressive test accuracy of 98.5%. However, the manuscript does not clearly assess the novelty and impact of this approach within the broader context of predictive modeling for insurance claims. Please compare the proposed model with other cutting-edge techniques in the field and discuss how it advances the current state of research. Additionally, highlighting the broader applicability of the model beyond the Egyptian insurance sector could amplify its impact.

b. While the conclusions are clear, they would benefit from a deeper discussion of the model's practical application. For instance, how could insurance companies incorporate this model into their decision-making processes? Also, further exploration of the model’s limitations (such as data biases and the lack of interpretability) would provide a more balanced perspective.

Additional comments

The manuscript presents a promising approach to insurance claim risk prediction but requires revisions. Clarify the novelty and impact of the proposed model, comparing it with existing techniques. Address ethical concerns, especially data privacy, and bias. Provide access to the code and dataset for reproducibility. Enhance the explanation of data pre processing, particularly regarding missing data and outliers. Expand on limitations and future directions, focusing on model interpretability and generalizability. Once these issues are addressed, the manuscript will be suitable for publication.

Annotated reviews are not available for download in order to protect the identity of reviewers who chose to remain anonymous.

---

## Round 0.2 · accepted · Accept

The paper was well improved. It can be accepted.

Reviewer 1 ·

Basic reporting

The authors have fixed previous concerns.

Experimental design

The authors have fixed previous concerns.

Validity of the findings

The authors have fixed previous concerns.

Additional comments

The authors have fixed previous concerns.

·

Basic reporting

.

Experimental design

.

Validity of the findings

.

Additional comments

.